# Study of Caspase 8 mutation in oral cancer and adjacent precancer tissues and implication in progression

Richa Singh[1¤], Shreya Das[2], Sila Datta[2], Anjana Mazumdar[2], Nidhan K. Biswas[3], Arindam Maitra[3], Partha P. Majumder[3], Sandip Ghose[2]*, Bidyut Roy[1]*

**1** Human Genetics Unit, Indian Statistical Institute, Kolkata, India, **2** Dr. R. Ahmed Dental College and Hospital, Kolkata, India, **3** National Institute of Biomedical Genomics, Kalyani, West Bengal, India

¤ Current address: Weill Cornell Medicine, Pathology and Laboratory Medicine, New York, New York, United States of America
* broy@isical.ac.in (BR); sanindra1967@gmail.com (SG)

**Data Availability Statement:** All relevant data are within the manuscript and Supporting Information files.

**Funding:** the authors received no specific funding for this work.

## Abstract

It is hypothesized that same driver gene mutations should be present in both oral leukoplakia and cancer tissues. So, we attempted to find out mutations at one of the driver genes, *CASP8*, in cancer and adjacent leukoplakia tissues. Patients (n = 27), affected by both of cancer and adjacent leukoplakia, were recruited for the study. Blood and tissue DNA samples were used to identify somatic mutations at *CASP8* by next generation sequencing method. In total, 56% (15 out of 27) cancer and 30% (8 out of 27) leukoplakia tissues had *CASP8* somatic mutations. In 8 patients, both cancer and adjacent leukoplakia tissues, located within 2–5 cm of tumor sites, had identical somatic mutations. But, in 7 patients, cancer samples had somatic mutations but none of the leukoplakia tissues, located beyond 5cm of tumor sites, had somatic mutations. Mutated allele frequencies at *CASP8* were found to be more in cancer compared to adjacent leukoplakia tissues. This study provides mutational evidence that oral cancer might have progressed from previously grown leukoplakia lesion. Leukoplakia tissues, located beyond 5cm of cancer sites, were free from mutation. The study implies that *CASP8* mutation could be one of the signatures for some of the leukoplakia to progress to oral cancer.

## Introduction

Among others, tobacco use, excessive consumption of alcohol and betel quid chewing are prominent risk factors especially, in South-east Asian countries. More than 300,000 new cases of oral cancer are diagnosed annually worldwide and the high incidence rates are observed in South and South-east Asia [1]. Oral cancers are commonly preceded by potentially premalignant oral epithelial lesions (PPOEL) such as leukoplakia, erythroplakia, submucous fibrosis and lichen planus [2], although majority of precancers do not progress to cancer. Malignant transformation rate of leukoplakia ranges from 0.13 to 34% [3–6]. Therefore, to control the

**Competing interests:** The authors have declared that no competing interests exist.

disease, it is important to examine mutations in leukoplakias and treat them to control their progression to cancer.

Evasion of apoptosis is regarded as one of the important hallmarks of cancer. This could be caused by the inactivation of *CASP8* (which encodes Caspase 8 protein) via multiple mechanisms such as mutations, epigenetic modifications, altered transcription, alternative splicing, post-translational changes. Studies, using next generation sequencing (NGS) method, reported that mutations in *CASP8* can vary from 7–34% in head and neck squamous cell carcinoma (HNSCC) across worldwide populations including India [7–10]. Based on Sanger sequencing method, another study on Southern Indian patient populations, reported 8% of oral cancer tissues had *CASP8* mutations but none in oral sub-mucous fibrosis tissues (another oral precancerous lesion) [11]. *CASP8* mutations with or without *FAT1* mutations have been used as an important criterion for classifying mutational profiles of oral cancer patients. Further, studies on oral cancer cell lines showed that inactivating mutation in *CASP8* can confer tumors with clonal growth advantage and increased cell migration [12].

Since the precancerous lesions are mostly asymptomatic and painless at the initial stages, patients are mostly reluctant to visit hospital making early diagnosis even more difficult. We were able to recruit patients into this study only when their lesions further aggravated. We recruited patients with both cancer and precancerous lesion fields and studied somatic mutations at *CASP8* in cancer as well as adjacent leukoplakia tissues by NGS method. We also compared the mutation data at *CASP8* between cancer and adjacent leukoplakia tissues, located at different distances from each other, to know whether the precancer and cancer tissues have common mutation which could be used as predictive signature for diagnosis at initial stage of progression.

## Materials and methods

### Collection of samples

This study had ethical approval from "Review committee for protection of research risk to humans" from Indian Statistical Institute, India and "Ethics committee" from Dr. R. Ahmed Dental College & Hospital, Kolkata, India. Written consents were taken from the patients with the information that their blood and tissues will be used for the present research. Patients, having both OSCC (oral squamous cell carcinoma) and adjacent leukoplakia were recruited between the year 2017(June) to 2018(May) from the hospital. Purposive and incidental type of sampling technique was used for recruitment of patients and all of them had tobacco smoking and/or chewing and/or alcohol consuming habit. Patients having verrucous carcinoma, submucosal fibrosis and lichen planus were not included in the study. International Cancer Genome Consortium (ICGC) reported (Data Portal; Tag: March 2010, E7: Study design and Statistical issues), that sample size of 500 tumors are necessary to detect somatic mutation at a cancer gene in 3% of the tumors. But in this pilot study, initially 40 patients were recruited within the stipulated time but 27 patients were found to be suitable for the study. Thirteen samples were excluded due to different reasons such as insufficient DNA from tissue, non-leukoplakia precancer and non-OSCC cancer after histopathological study, insufficient data after NGS *etc*. From each patient, 2ml venous blood and biopsy punch from tumor and leukoplakia tissues were collected. Blood samples were kept at -20˚C until extraction of DNA. Part of the leukoplakia and cancer tissue samples were kept in *RNA Later* solution and stored at -20˚C until extraction of DNA. Remaining part of the tissue samples were used for histopathological examination.

### DNA isolation, library preparation and sequencing

DNA was isolated from leukoplakia, cancer and blood tissues using QIAGEN DNA isolation kit. Concentration and quality of DNA was checked by *Nanodrop 2000* (Thermo Scientific). From isolated DNA, library for NGS was prepared by Illumina library Preparation Kit and checked by Kapa quantification kit (Kapa Biosystems, Wilmington, MA, USA). Sequencing was performed using Illumina HiSeq 2500 paired-end 100bp protocol.

### Variant calling and bioinformatics analysis

Somatic mutations were identified in tumor and leukoplakia tissues by comparing it with blood from same patient using *MuTect2*[13] and all variants were annotated using *ANNOVAR* [14] (S1 Data). Mutation frequencies were verified from *mpileup* files, generated by *Samtools* for each sample[15]. Public databases like *TCGA* (The Caner Genome Atlas), *COSMIC* (Catalogue of Somatic Mutation in Cancer) and *ClinVar* [16] annotations were used to identify reported and novel variants.

### Statistical analysis

For somatic mutation calling, *Mutect2* used Bayesian somatic genotyping model which detected mutation even at very low allele frequency and allelic fraction showing high specificity [13]. The allelic frequencies obtained from *Mutect2* of each leukoplakia and tumor tissues, were compared with frequency in respective blood samples. Significant difference (i.e. p-value <0.05) in mutation frequencies between blood and disease tissues was calculated by Fisher's exact test.

## Results

### Demography

The study included 27 patients affected by both OSCC and adjacent leukoplakia and having history of consuming tobacco either in chewable or/and smoking form. Patients included both males (n = 21) and females (n = 6) and age of the patients ranged from 32 to 70 years (Table 1). All tumors, except one which was moderately differentiated, were well differentiated OSCC. Among all leukoplakia tissues, 13, 11 and 3 samples were diagnosed as mild, moderate and severe degree of dysplasia, respectively (Table 1). Some of the patients (n = 11) had a margin of >5cm between OSCC and leukoplakia tissues. Remaining patients (n = 16) had a margin of 2-5cm between OSCC and leukoplakia tissues.

### Mutations in *CASP8*

At 100x sequencing depth, mutations in *CASP8* were identified in 56% of tumor (i.e. 15/27) and 30% of leukoplakia (i.e. 8/27) tissues. Among the 15 patients, 8 had identical *CASP8* mutations in both tumor and leukoplakia lesions (located within 2–5 cm of cancer tissue) (Table 2, S1 Data). Remaining 7 patients had somatic mutation in *CASP8* in tumor but not in its adjacent leukoplakia tissues (located either within 2–5 cm or beyond 5 cm of cancer tissues) (Table 3). All these 15 mutations were distributed in exons between 2 and 8 which codes for death effector domain (DED), Caspase domain (Peptidase_C14) and linker regions (Fig 1A). *CASP8* mutations, reported previously in other HNSCC samples from TCGA cohorts, showed mutations distributed across all domains of the protein (Fig 1B). Although we did not have survival and follow-up history for the patients used in this study, we did observe that mutations in *CASP8* in TCGA HNSCC samples showed a significant decrease in overall survival (Log Rank test P Value of 8.927E$^{-04}$) (Fig 1C) [17, 18]. Among 8 somatic mutations present in

**Table 1. Demography of patients with tobacco habits and histopathological observations (n = 27).**

| Patient ID | Age (years) | Sex | Habit | | | Histopathological subtype | | Tumor Size |
|---|---|---|---|---|---|---|---|---|
| | | | Tobacco smoking | Tobacco chewing | Alcohol consumption | Tumor | Leukoplakia | |
| RADS2 | 53 | M | Y | Y | - | MDSCC | MILD DYSPLASIA | T2 |
| RADS3 | 41 | F | - | Y | - | WDSCC | MODERATE DYSPLASIA | T1 |
| RADS4 | 32 | M | - | Y | Y | WDSCC | MILD DYSPLASIA | T4a |
| RADS5 | 50 | F | - | Y | - | WDSCC | MILD DYSPLASIA | T4a |
| RADS6 | 60 | M | Y | - | - | WDSCC | MODERATE DYSPLASIA | T1 |
| RADS7 | 65 | M | Y | - | Y | WDSCC | MODERATE DYSPLASIA | T2 |
| RADS8 | 54 | M | Y | - | - | WDSCC | MODERATE DYSPLASIA | T4a |
| RADS10 | 65 | M | - | Y | Y | WDSCC | MODERATE DYSPLASIA | T4a |
| RADS13 | 48 | F | - | Y | - | WDSCC | MILD DYSPLASIA | T4a |
| RADS17 | 53 | M | Y | Y | - | WDSCC | MILD DYSPLASIA | T4a |
| RADS18 | 58 | M | Y | Y | - | WDSCC | MILD DYSPLASIA | T4a |
| RADS19 | 41 | M | - | Y | - | WDSCC | SEVERE DYSPLASIA | T4a |
| RADS23 | 45 | M | - | Y | Y | WDSCC | MILD DYSPLASIA | T4a |
| RADS27 | 50 | M | Y | Y | - | WDSCC | MILD DYSPLASIA | T4a |
| RADS28 | 55 | M | Y | Y | - | WDSCC | SEVERE DYSPLASIA | T4a |
| RADS29 | 36 | M | - | Y | - | WDSCC | MODERATE DYSPLASIA | T1 |
| RADS33 | 45 | F | - | Y | - | WDSCC | SEVERE DYSPLASIA | T2 |
| RADS34 | 65 | M | - | Y | - | WDSCC | MILD DYSPLASIA | T3 |
| RADS36 | 39 | M | Y | Y | - | WDSCC | MILD DYSPLASIA | T4a |
| RADS37 | 55 | M | - | Y | - | WDSCC | MILD DYSPLASIA | T1 |
| RADS39 | 45 | F | - | Y | - | WDSCC | MODERATE DYSPLASIA | T1 |
| RADS40 | 47 | M | - | Y | - | WDSCC | MODERATE DYSPLASIA | T1 |
| RADS42 | 53 | M | Y | - | Y | WDSCC | MODERATE DYSPLASIA | T1 |
| RADS43 | 70 | M | - | Y | - | WDSCC | MODERATE DYSPLASIA | T4a |
| RADS45 | 60 | M | Y | Y | - | WDSCC | MODERATE DYSPLASIA | T1 |
| RADS47 | 65 | F | - | Y | - | WDSCC | MILD DYSPLASIA | T4a |
| RAD52 | 61 | M | - | Y | - | WDSCC | MILD DYSPLASIA | T1 |

[MDSCC and WDSCC: moderately and well differentiated squamous cell carcinoma;

M—Male; F—Female; Y—Yes]

**Table 2. Presence of mutation in *CASP8* in both of cancer and adjacent leukoplakia tissues from same patient (n = 8).**

| Sample ID | Mutation in both of cancer and adjacent leukoplakia tissues [*] | Tumor size |
|---|---|---|
| RADS 3 | L428Q; T>A; nonsynonymous; exon 8 | T1 |
| RADS 4 | Q225X; C>T; stop gain; exon 4 | T4a |
| RADS5 | G310D; G>A; nonsynonymous; exon7 | T4a |
| RADS 10 | R417X; C>T; stop gain; exon 7 | T4a |
| RADS27 | E204X; G>T; stop gain; exon 4 | T4a |
| RADS28 | D200fs; del TATT; frameshift deletion; exon 4 | T4a |
| RADS37 | R218Q; G>A; Nonsynonymous; exon 6 | T1 |
| RADS47 | T258fs; C>CT; frameshift due to insertion, exon 7 | T4a |

[*] all leukoplakia tissues were located within 2–5 cm from tumor

**Table 3. Presence of mutation in *CASP8* in cancer but not in adjacent leukoplakia tissues from same patients (n = 7).**

| Sample ID | Mutation in leukoplakia | Mutation in tumor |
|---|---|---|
| $ RADS7 | No mutation | R398X; C>T; stop gain; exon 7 |
| $ RADS8 | No mutation | I296S; T>G; nonsynonymous; exon 7 |
| * RADS17 | No mutation | Q150X; C>T; stop gain; exon 2 |
| * RADS18 | No mutation | D293V; A>T; nonsynonymous; exon 7 |
| * RADS23 | No mutation | P379L; C>T; nonsynonymous; exon 7 |
| $ RADS36 | No mutation | *Y160fs; del T*; frameshift deletion; exon 2 |
| $ RADS39 | No mutation | E434X; G>T; stop gain; exon8 |

$: leukoplakias were located 5cm away from tumors

*: leukoplakias were located within 2-5cm of tumors

both of tumor and adjacent leukoplakia tissues (i.e. located within 2-5cm of tumor site), 3 were non-synonymous, 2 were frame-shift and remaining 3 were stop-gain mutations (Table 2).

No mutated allele at *CASP8* was detected in blood DNA indicating that blood DNA was not contaminated with tissue DNA. The frequency of mutated alleles increased progressively from leukoplakia to cancer tissues in 7 patients (Fig 2). In most of the samples, mutation frequency in leukoplakia versus blood and cancer versus leukoplakia were significantly different (p value < 0.05, Fisher's exact test). In few samples, changes in mutated allele frequencies in cancer versus leukoplakia (e.g. RADS3) and leukoplakia versus blood (e.g. RADS27 and RADS47) were not significant.

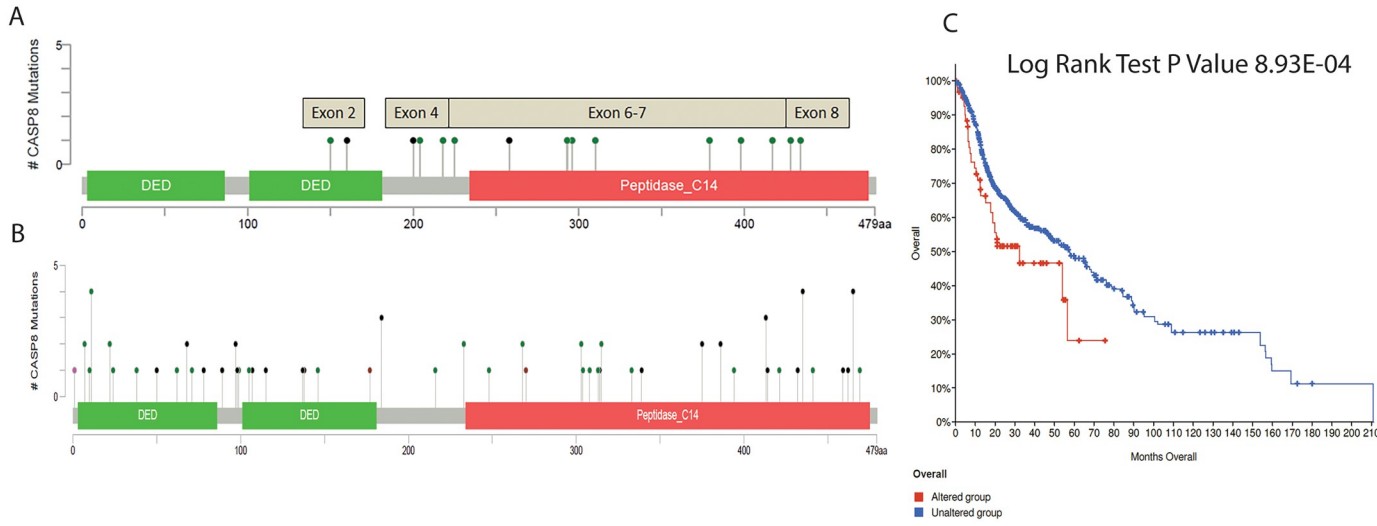

**Fig 1. Schematic diagram of distribution of somatic mutations in *CASP8* over different exons and protein domains. A**. Somatic mutations (n = 15) were shown at different exons of *CASP8* in the studied samples of oral cancer. Twelve green headed lines denote non-synonymous and stop-gain mutations. Three black headed lines denote frame shift mutations. **B**. Somatic mutations reported in HNSCC samples from TCGA cohorts. **C**. Overall survival rate of patients with and without *CASP8* mutations in HNSCC samples from TCGA cohorts.

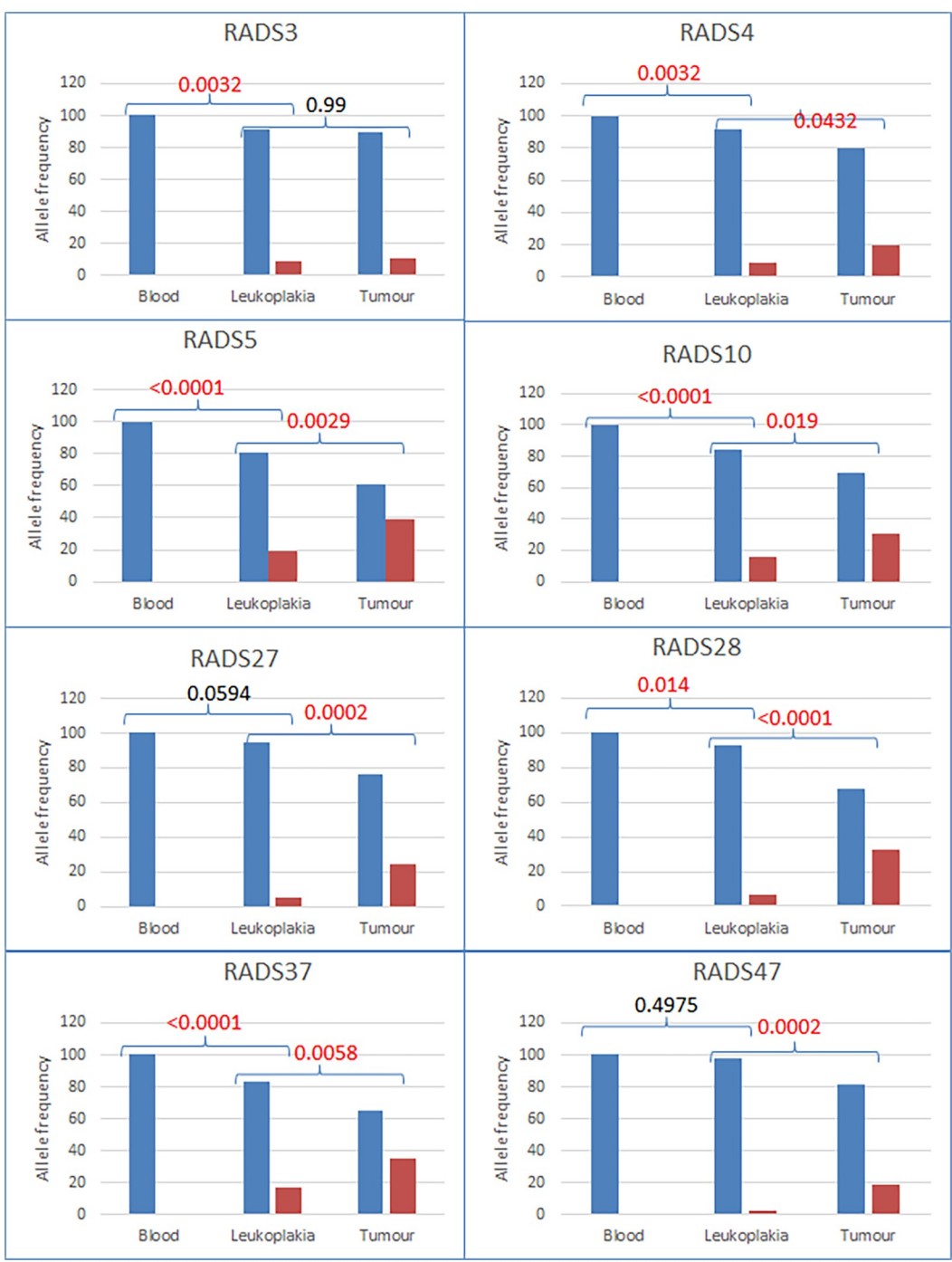

**Fig 2. Wild type and mutated allele frequencies of *CASP8* in blood, leukoplakia and tumor tissues of the patients.** Wild type (blue bars) in blood DNA and mutated allele (red bars) frequencies in adjacent (located within 2–5 cm of tumor sites) leukoplakia and cancer tissues were plotted. Both leukoplakia and cancer tissues had identical type of mutation. For each of the plotted samples, mutated allele was found to be absent in blood DNA but present in leukoplakia and tumor tissues. Its frequency increased in cancer compared to adjacent leukoplakia tissue (p <0.05 for seven samples while p>0.05 in one sample).

### Novel *CASP8* mutations and its functional impact

Amongst the 15 somatic mutations, 5 were reported previously in various cancers described in *TCGA* data base (http://www.cancer.gov/tcga) [17, 18]. Two of these mutations were reported in HNSCC while others were reported in cancer of cervix uteri, bladder, stomach and colon [17, 18]. These two mutations reported in HNSCC (*R398X* and *R417X*) were predicted to have high impact by variant effect prediction tools while the remaining 3 mutations (*R218Q*, *G310D* and *P379L*) were missense and predicted to have deleterious and probably damaging effect by SIFT and Polyphen, respectively [17, 18]. The high impact mutations (*R398X* and *R417X*) were stop-gain and one of them, *R417X* (in RADS10), was also present in both and adjacent leukoplakia tissue. Two other known missense mutations (*R218Q* and *G310D*), mentioned previously, were also present in both tumor and adjacent leukoplakia. Remaining 10 mutations, identified in this study are novel. Amongst these novel mutations, 5 mutations (*D200fs*, *E204X*, *Q225X*, *T258fs* and *L428Q*) were present in both tumor and adjacent leukoplakia tissues. All these 5 mutations, except one, were loss-of function truncating mutations with either stop-gain or frame-shift effects. Other 5 novel mutations tumors, not identified in adjacent leukoplakia tissues, included missense, frame-shift and stop-gain alterations.

## Discussion

To the best of our knowledge, this is the first study showing presence of *CASP8* mutation in oral cancer as well as adjacent leukoplakia tissues from same patients. With this small sample size, 56% of the tumors (i.e. 15/27 tumors) had *CASP8* mutations which supports the observations reported in other studies [7–10]. Adjacent leukoplakias were located either within 2-5cm or >5cm away from cancer tissues. We observed that some (8/16 i.e. 50%) of the adjacent leukoplakia tissues (located within 2–5 cm of the tumor tissues) had identical somatic mutations like tumor tissues (Table 2). For these tumors, it may be interpreted that *CASP8* mutation occurred in an early stage of the lesion, i.e. in a leukoplakia patch, where few cells acquired more mutations in other genes and were transformed into cancer. In a study on cancer model, it was shown that sequential accumulation of mutations (such as three mutations in tumor suppressors or oncogenes) are necessary to develop tumors [19]. In support of this, we hypothesize, that *CASP8* mutations in these leukoplakia tissues, may be an early event followed by mutations in other driver genes which help in progression to malignant transformation.

Although three other leukoplakia tissues were located within 2–5 cm of the tumor tissues (Table 3), they did not show *CASP8* mutations but respective tumor tissues had mutations. This led us to interpret that *CASP8* mutations were late stage drivers in these tumors and acquired after precancer stage. It may be interpreted that these mutations, observed in only tumor tissues but not in nearby leukoplakias (within 2–5 cm), may have different oncogenic potential than other *CASP8* mutations. Previous report of different levels of oncogenic potential of variants from the same gene has been well elucidated [20]. It is interesting to note that four other tumors had *CASP8* mutations but the leukoplakias (located at >5 cm away from tumor tissues) did not have mutation (Table 3). We hypothesized that these cancers and leukoplakia tissues might have grown independently from normal epithelium cells and the tumor acquired *CASP8* mutations during progression. So, it may be suggested that a margin of 5 cm around tumor can be regarded as safe margin to avoid leukoplakia patch during onco-surgery [21]. Other issues, such as field of cancerization should be studied using molecular genetic markers to authenticate or negate this model of progression of cancer from existing leukoplakia tissue. Studies on other cancer model also highlighted the importance of studying field of cancerization [22].

We did not find any correlation between occurrence of *CASP8* mutation and gradation of dysplastic nature of 8leukoplakia tissues. Five of them were mild, 2 were moderate and 1 was severely dysplastic in histopathological appearances (Tables 1 and 2). Based on the results, we inferred that some molecular changes such as *CASP8* mutations may not always get reflected in the histopathological features which are regarded as basis for grading the dysplastic nature of leukoplakia. Fifteen tumors, with *CASP8* mutations, were moderate/well differentiated carcinoma and clinically $N_0$ status but had different tumor sizes (such as $T_1$, $T_2$, $T_3$ and $T_{4a}$) (Tables 1 and 2). These hispathological findings and mutation data together suggest that *CASP8* mutation may even occur at initial stage of carcinogenesis.

Using Sanger sequencing method, a study reported absence of *CASP8* mutation in oral cancer samples collected from South Indian patients [23]. But another study, using same method on a different cohort from South India, reported that 8% of cancer patients had somatic mutations at *CASP8* but none of the oral sub mucous fibrosis samples (one of the oral precancers) showed mutation [11]. It is to be noted that these two studies used Sanger sequencing method which is not suitable to detect low frequency somatic mutation in contrast to NGS method applied in this study (Fig 2).

It was also observed that mutated allele frequencies increased significantly in leukoplakia and adjacent cancer tissues compared to blood DNA from same patients. More interestingly, mutated allele frequencies in most of the tumor tissues were also significantly higher compared to those in adjacent leukoplakia tissues (Fig 2). This suggests that cells with *CASP8* mutations might have gained selective advantage during progression from leukoplakia to tumor.

Different types of mutations in *CASP8* have been detected in different cancer tissues such as HNSCC [7–10], hepatocellular [24], gastric [25], and colorectal carcinomas [26]. These mutations include missense, nonsense and frame shift mutations which might lead to diversity in tumorigenesis, although experimental evidences are needed. *CASP8* is often found to be mutated in different frequencies (10–34%) in HNSCC from different patient populations [7–10] and could be considered as a central player in the extrinsic apoptotic cascade triggered by death receptors stimulation. Here, we found that *CASP8* mutation may emerge as a potential signature for progression of oral cancer from leukoplakia. It may be noted that although clinically diagnosed leukoplakia did not show any sign of invasion but these were not free from molecular alteration. So, clinically diagnosed leukoplakia should be included for mutation study and subsequent surgery to prevent progression to malignancy.

## Supporting information

**S1 Data.**
(XLSX)

## Acknowledgments

Authors like to acknowledge support and help of the patients for providing blood and tissue samples for the study.

## Author Contributions

**Conceptualization:** Bidyut Roy.

**Formal analysis:** Richa Singh.

**Funding acquisition:** Bidyut Roy.

**Investigation:** Richa Singh, Shreya Das, Sila Datta, Anjana Mazumdar, Sandip Ghose.

**Methodology:** Arindam Maitra, Partha P. Majumder.

**Software:** Richa Singh.

**Supervision:** Bidyut Roy.

**Validation:** Nidhan K. Biswas.

**Visualization:** Partha P. Majumder, Sandip Ghose.

**Writing – original draft:** Richa Singh.

**Writing – review & editing:** Bidyut Roy.

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
