## [Decision Letter · Decision Letter 0]

19 Mar 2020

PONE-D-20-04461

Study of Caspase 8 mutation in oral cancer and adjacent precancer tissues and implication in progression

PLOS ONE

Dear Dr. Roy,

Thank you for submitting your manuscript to PLOS ONE. After careful consideration, we feel that it has merit but does not fully meet PLOS ONE’s publication criteria as it currently stands. Therefore, we invite you to submit a revised version of the manuscript that addresses the points raised during the review process.

We would appreciate receiving your revised manuscript by May 03 2020 11:59PM. To enhance the reproducibility of your results, we recommend that if applicable you deposit your laboratory protocols in protocols.io, where a protocol can be assigned its own identifier (DOI) such that it can be cited independently in the future. For instructions see: http://journals.plos.org/plosone/s/submission-guidelines#loc-laboratory-protocols

We look forward to receiving your revised manuscript.

Kind regards,

Alvaro Galli

Academic Editor

PLOS ONE

Journal Requirements:

2.Thank you for including your ethics statement: 'This study had ethical approval from Indian Statistical Institute and Dr. R. Ahmed Dental College & Hospital, Kolkata, India.'

a.Please amend your current ethics statement to include the full name of the ethics committee/institutional review board(s) that approved your specific study.  

b.Once you have amended this/these statement(s) in the Methods section of the manuscript, please add the same text to the “Ethics Statement” field of the submission form (via “Edit Submission”).

For additional information about PLOS ONE ethical requirements for human subjects research, please refer to " ext-link-type="uri" xlink:type="simple">http://journals.plos.org/plosone/s/submission-guidelines#loc-human-subjects-research."

3. Please provide additional details regarding participant consent. In the ethics statement in the Methods and online submission information, please ensure that you have specified whether consent was informed.

4. To comply with PLOS ONE submission guidelines, in your Methods section, please provide additional information regarding your statistical analyses. For more information on PLOS ONE's expectations for statistical reporting, please see https://journals.plos.org/plosone/s/submission-guidelines.#loc-statistical-reporting.

5. In your Methods section, please provide additional information about the participant recruitment method and the demographic details of your participants. Please ensure you have provided sufficient details to replicate the analyses such as: a) the recruitment date range (month and year), b) a description of how participants were recruited, and c) descriptions of where participants were recruited (name of the hospital from which participants were recruited.

6. Please provide a sample size and power calculation in the Methods, or discuss the reasons for not performing one before study initiation.

7. In your Data Availability statement, you have not specified where the minimal data set underlying the results described in your manuscript can be found. PLOS defines a study's minimal data set as the underlying data used to reach the conclusions drawn in the manuscript and any additional data required to replicate the reported study findings in their entirety. All PLOS journals require that the minimal data set be made fully available. For more information about our data policy, please see http://journals.plos.org/plosone/s/data-availability.

8. We note that you have included the phrase “data not shown” in your manuscript. Unfortunately, this does not meet our data sharing requirements. PLOS does not permit references to inaccessible data. We require that authors provide all relevant data within the paper, Supporting Information files, or in an acceptable, public repository. Please add a citation to support this phrase or upload the data that corresponds with these findings to a stable repository (such as Figshare or Dryad) and provide and URLs, DOIs, or accession numbers that may be used to access these data. Or, if the data are not a core part of the research being presented in your study, we ask that you remove the phrase that refers to these data.

Reviewers' comments:

Reviewer's Responses to Questions

**Comments to the Author**

1. Is the manuscript technically sound, and do the data support the conclusions?

Reviewer #1: Yes

Reviewer #2: Yes

2. Has the statistical analysis been performed appropriately and rigorously? 

Reviewer #1: Yes

Reviewer #2: Yes

3. Have the authors made all data underlying the findings in their manuscript fully available?

Reviewer #1: Yes

Reviewer #2: Yes

4. Is the manuscript presented in an intelligible fashion and written in standard English?

Reviewer #1: No

Reviewer #2: Yes

5. Review Comments to the Author

Reviewer #1: The goal of the study was to determine whether Caspase 8 mutations were present in early lesions and cancers in the oral cavity. Adjacent leukoplakia and cancers were analyzed from 27 patients. DNA was isolated and sequenced using Next Generation sequencing, however, only mutations in Caspase 8 were assessed here. In this study 30% of leukoplakia and 56% of cancers had mutations in Caspase 8. In 8 cases, the same somatic mutations was observed in the leukoplakia as the cancer. The study is rather limited in sample number but the data largely support the conclusions that are drawn. Several concerns need to be addressed.

Major

1) The authors argue that Caspase 8 is a driver of tumorigenesis in the oral cavity. What is the evidence that this is the case. Previous studies have found that up to 34% of HNSCC carry mutations in Caspase 8. If Caspase 8 is a driver, it must be a driver in only some cancers. Whether Caspase 8 is a driver or not needs to be discussed in more detail.

2) In the Dscussion, the authors argue in Lines 191-194 that more mutations are necessary to drive transformation. If this is correct, is Caspae 8 really a driver. Is it necessary, sufficient, both, or neither.

2) Have any animal models been generated to study the role of Caspae 8 in HNSCC?

3) Need to describe the exclusion criteria for the study.

4) Need to explain specifically why 13 patients were dropped.

5) In the Discussion, the authors state in lines 208-210 that "molecular changes are not always reflected in histopathological features". This statement needs to be clarified or justified. Are they referring to Caspase 8 or other genes?

6) Dr. Liskay and his laboratory staff demonstrated that a field of Apc-deficient cells was critical to the establishment of a tumor in the colon using elegant mouse models (Carcinogenesis 35:237, 2014). In fact, they demonstrated the size of the field is critical. This finding support statement in lines 230-232 in the Discussion.

7) Recent studies have demonstrated that all mutant alleles in cancer drivers are not created equal, e.g., different mutations in KRAS have different degrees of oncogenic potential (Nat Commun 8: 2053, 2017). Seems that this point should be discussed because the authors present information regarding the different mutations observed in Caspase 8.

Minor

1) Tables 1 and 2 could easily be combined into one.

2) Table 3 does not need to restate the shared mutation in columns 3 and 4.

3) The grammar needs to be corrected throughout the manuscript.

Reviewer #2: This study reports on the investigation if CASP8 mutations which are common in oral cancers, are also present in pre-neoplastic lesions, and as such investigate whether CASP8 mutations are an early event in the transition to oral cancer.

Does figure 1 show the mutations of the patients reported in this report? If so make this clear in the figure caption.

6. PLOS authors have the option to publish the peer review history of their article (what does this mean?). If published, this will include your full peer review and any attached files.

Reviewer #1: No

Reviewer #2: No

---

## [Author Response · Author response to Decision Letter 0]

22 Apr 2020

Editor’s note:

1. Cover letter has been modified

2. Ethical Statement has been updated in Method and submission form

3. Participant consent has been updated in method and submission form

4. Additional information on statistical method has been updated in method

5. Participant recruitment has been elaborated in method

6. Sample size has been explained in method

7. Data availability has been mentioned and uploaded

8. “Data not shown” has been corrected

Reviewers’ note:

We like to thank reviewers for their excellent comments to improve the manuscript. All comments have been addressed.

Response to Reviewers

1) The authors argue that Caspase 8 is a driver of tumorigenesis in the oral cavity. What is the evidence that this is the case. Previous studies have found that up to 34% of HNSCC carry mutations in Caspase 8. If Caspase 8 is a driver, it must be a driver in only some cancers. 

Authors’ response: Role of CASP8 has been elucidated in lines 63-67. Since, inactivating mutations of CASP8 gives tumor growth advantage, we do expect to see a higher frequency of mutated gene in cancer.

We have shown that CASP8 may have worked as driver mutation in some of the leukoplakia and tumor tissues (lines 215-224). 

2) In the Discussion, the authors argue in Lines 191-194 that more mutations are necessary to drive transformation. If this is correct, is Caspase 8 really a driver. Is it necessary, sufficient, both, or neither.

Authors’ response: In a study by Tomasetti et al, PNAS, 2015(PMID 25535351), the authors showed that sequential accumulation of mutations (that is, three mutations in tumor suppressors or oncogenes) are necessary to develop tumors. Along these lines we hypothesize, that mutations in CASP8 may be an early event in some of the leukoplakias followed by mutations in other driver genes which then lead to tumorigenesis. This is further elaborated in the manuscript for clarity (lines 215-224). 

3) Have any animal models been generated to study the role of Caspase 8 in HNSCC?

Authors’ response: We found reports on CASP8-deficient mice models of colorectal cancer but not for HNSCC. But study on cell lines with CASP8 mutation has been reported and mentioned in text (lines 65-67).

4) Need to describe the exclusion criteria for the study.Need to explain specifically why 13 patients were dropped.

Authors’ response: These queries have been mentioned in lines 92-94.

5) In the Discussion, the authors state in lines 208-210 that “molecular changes are not always reflected in histopathological features”. This statement needs to be clarified or justified. Are they referring to Caspase 8 or other genes?

Authors’ comments: The authors here refer to molecular changes such as CASP8 mutations which may not show distinct phenotype change in histopathological features of leukoplakia and may not show direct correlation with grades of lesion severity (Lines 242-251). 

6) Dr. Liskay and his laboratory staff demonstrated that a field of Apc-deficient cells was critical to the establishment of a tumor in the colon using elegant mouse models (Carcinogenesis 35:237, 2014). In fact, they demonstrated the size of the field is critical. This finding support statement in lines 230-232 in the Discussion.

Authors’ response: This excellent observation has been mentioned in manuscript with reference (lines 238-241). 

7) Recent studies have demonstrated that all mutant alleles in cancer drivers are not created equal, e.g., different mutations in KRAS have different degrees of oncogenic potential (Nat Commun 8: 2053, 2017). Seems that this point should be discussed because the authors present information regarding the different mutations observed in Caspase 8

Authors’ response: This important observation has been mentioned in discussion section (lines 228-231, 366-268).

Minor

1) Tables 1 and 2 could easily be combined into one.

Authors’ response: Table 1 and 2 are combined as per reviewers’ suggestion.

2) Table 3 does not need to restate the shared mutation in columns 3 and 4.

Author’s response: Table is modified as per reviewers’ suggestion.

3) The grammar needs to be corrected throughout the manuscript.

Author’s response: We have re-written many sections of the manuscript to improvise the language.

Reviewer #2: This study reports on the investigation if CASP8 mutations which are common in oral cancers, are also present in pre-neoplastic lesions, and as such investigate whether CASP8 mutations are an early event in the transition to oral cancer.

Does figure 1 show the mutations of the patients reported in this report? If so make this clear in the figure caption.

Author’s response: Yes, mutations shown in Figure 1A were observed in 15 of the tumors included in this study. To compare it with mutation present in other HNSCC patients’ samples of TCGA cohorts, we have included Figure 1B. Figure 1B shows CASP8 mutations reported in a TCGA samples. Figure caption has been changed accordingly.

---

## [Decision Letter · Decision Letter 1]

28 Apr 2020

Study of Caspase 8 mutation in oral cancer and adjacent precancer tissues and implication in progression

PONE-D-20-04461R1

Dear Dr. Roy,

We are pleased to inform you that your manuscript has been judged scientifically suitable for publication and will be formally accepted for publication once it complies with all outstanding technical requirements.

With kind regards,

Alvaro Galli

Academic Editor

PLOS ONE

Additional Editor Comments (optional):

Reviewers' comments:

Reviewer's Responses to Questions

**Comments to the Author**

1. If the authors have adequately addressed your comments raised in a previous round of review and you feel that this manuscript is now acceptable for publication, you may indicate that here to bypass the “Comments to the Author” section, enter your conflict of interest statement in the “Confidential to Editor” section, and submit your "Accept" recommendation.

Reviewer #1: All comments have been addressed

2. Is the manuscript technically sound, and do the data support the conclusions?

Reviewer #1: Yes

3. Has the statistical analysis been performed appropriately and rigorously? 

Reviewer #1: Yes

4. Have the authors made all data underlying the findings in their manuscript fully available?

Reviewer #1: Yes

5. Is the manuscript presented in an intelligible fashion and written in standard English?

Reviewer #1: Yes

6. Review Comments to the Author

Reviewer #1: A comments have been addressed. The writing still needs some editing.

Driver is spelled as diver in Abstract.

7. PLOS authors have the option to publish the peer review history of their article (what does this mean?). If published, this will include your full peer review and any attached files.

Reviewer #1: No

---

## [Editor Report · Acceptance letter]

22 May 2020

PONE-D-20-04461R1 

Study of Caspase 8 mutation in oral cancer and adjacent precancer tissues and implication in progression 

Dear Dr. Roy:

I am pleased to inform you that your manuscript has been deemed suitable for publication in PLOS ONE. Congratulations! Your manuscript is now with our production department. 

With kind regards,

on behalf of

Dr. Alvaro Galli 

Academic Editor

PLOS ONE